# Separating Infectious Proctitis from Inflammatory Bowel Disease—A Common Clinical Conundrum

**DOI:** 10.3390/microorganisms12122395

**Published:** 2024-11-22

**Authors:** Richard Hall, Kamal Patel, Andrew Poullis, Richard Pollok, Sailish Honap

**Affiliations:** 1Department of Gastroenterology, St George’s University Hospital, London SW17 0QT, UK; richard.hall@stgeorges.nhs.uk (R.H.);; 2Institute of Infection and Immunity, St George’s University, London SW17 0RE, UK; 3School of Immunology and Microbial Sciences, King’s College London, London SE1 9NH, UK

**Keywords:** proctitis, inflammatory bowel disease, sexually transmitted infection, lymphogranuloma venereum, *Chlamydia trachomatis*, *Neisseria gonorrhoeae*, Mpox

## Abstract

Proctitis refers to inflammation in the rectum and may result in rectal bleeding, discharge, urgency, tenesmus, and lower abdominal pain. It is a common presentation, particularly in genitourinary medicine and gastroenterology, as the two most common causes are sexually transmitted infections and inflammatory bowel disease. The incidence of infective proctitis is rising, particularly amongst high-risk groups, including men who have sex with men, those with HIV seropositive status, and those participating in high-risk sexual behaviours. The most commonly isolated organisms are *Neisseria gonorrhoeae*, *Chlamydia trachomatis*, *Treponema palladium*, herpes simplex virus, and *Mycoplasma genitalium*. Recently, proctitis was also identified as a common feature during the Mpox outbreak. Distinguishing infective proctitis from inflammatory bowel disease remains a significant clinical challenge as there is significant overlap in the clinical presentation and their endoscopic and histological features. This review compares and highlights the distinguishing hallmarks of both inflammatory and infective causes of proctitis. It provides a practical guide to describe the key features that clinicians should focus on in both clinical and key diagnostic investigations to avoid potential misdiagnosis.

## 1. Introduction

Proctitis is defined as inflammation of the rectal mucosa, distal to the rectosigmoid junction and within 20 cm of the anal verge. The symptoms of proctitis include rectal bleeding, mucus discharge, tenesmus, proctalgia, faecal incontinence, constipation, urgency, incomplete defecation, fever, and abdominal pain [1]. The two most common causes of patients presenting with proctitis are infective causes and inflammatory bowel disease (IBD). Whilst there is a broader differential diagnosis, including radiation proctitis, traumatic proctitis, malignancy, solitary rectal ulcer, diversion colitis, and ischaemia, distinguishing between IBD and infective causes provides the biggest clinical challenge as they share many overlapping clinical, endoscopic, histological, and radiological features.

Infective proctitis is usually caused by sexually transmitted infections. Globally, the most commonly isolated organisms are *Neisseria gonorrhoeae*, *Chlamydia trachomatis*, and *Treponema palladium*. Incidences of these infections have been increasing globally [2] and are rising, particularly in the highest-risk groups, including men who have sex with men (MSM), those with HIV seropositive status, and individuals performing receptive anal intercourse (see Table 1). The recent introduction and availability of pre-exposure prophylaxis (PrEP) has led to a further increase in infection rates amongst these high-risk groups [3]. Proctitis has been identified as a common symptom in the recent Mpox outbreak [4].

Inflammatory proctitis refers to patients diagnosed with inflammatory bowel disease (IBD), and there are two main conditions: ulcerative colitis (UC) and Crohn’s disease (CD). The worldwide incidence of inflammatory bowel disease is increasing [5,6] and is expected to continue to rise, particularly in middle- and low-income countries. Up to 30% of newly diagnosed patients with ulcerative colitis present with isolated proctitis [7].

As the cases of both infective and inflammatory proctitis increase globally, distinguishing between these will become an increasingly common clinical conundrum. Making an incorrect diagnosis can lead to incorrect and/or delayed treatments, leading to complications that could otherwise be avoided [8]. The aim of this review is to identify the key features of inflammatory and infective proctitis, and to highlight the distinguishing features to assist clinicians in the diagnostic process.

## 2. Inflammatory Bowel Disease

Inflammatory bowel disease is a chronic immune-mediated disease affecting the gastrointestinal tract. It typically presents between the 2nd and 4th decades of life and is broadly sub-divided into UC [9] and CD [10]. The pathogenesis and pathophysiology are incompletely understood but are thought to involve a complex interplay between environmental factors, immune dysregulation, dysbiosis, and genetic predisposition [11].

IBD was historically considered a disease mainly of high-income countries. In North America and Europe, the prevalence was previously reported to be at least 0.3% [6] but is now approaching 1% [5]. As countries become increasingly industrialised, there is an established trend of increasing IBD prevalence in middle- and low-income countries [12]. The incidence is similar between men and women; there is no difference between biological sex in CD, with some studies suggesting marginally higher rates of UC amongst men compared to women [13].

Currently, there is no single diagnostic test for IBD, with clinicians using a combination of clinical history, examination, biomarkers, radiology, endoscopy, and histopathology to make a diagnosis.

### 2.1. Ulcerative Proctitis

UC is a chronic inflammatory bowel disease characterised by rectal bleeding, increased bowel frequency, diarrhoea, tenesmus, and urgency [14]. Typically, UC affects the rectum and extends in a continuous and circumferential manner proximally to a variable degree. UC is commonly classified according to the extent of mucosal involvement according to the Montreal classification [15], ranging from proctitis to left-sided colitis or extensive colitis. Ulcerative proctitis is defined as inflammation confined to the rectum (distal 20 cm from the anal verge). In Western populations, between 25 and 35% of patients newly diagnosed with UC present with isolated ulcerative proctitis [7]. Eastern populations have reported higher rates of proctitis on presentation [16,17]. Despite having limited disease extent and lower rates of hospitalisation, colorectal cancer, and colectomy [7], patients with ulcerative proctitis carry a high symptom burden. The most commonly reported symptoms in patients presenting with ulcerative proctitis are bloody diarrhoea and rectal urgency, which can be very distressing and compromise the quality of life [18]. Other symptoms include tenesmus, faecal incontinence, and, in 5–10% of patients, constipation. Despite the significant symptom burden, the onset of UC is usually insidious [9], and, consequently, symptoms are present for a considerable period of time before a diagnosis is made [19].

In UC, endoscopic inflammatory changes commence from the anal verge and extend in a proximal and continuous fashion with an abrupt demarcation to normal mucosa [20]. Features of inflammation include mucosal erythema, mucosal oedema, obliteration of vascular pattern, mucosal or luminal bleeding, and the presence of ulcers or erosions embedded in inflamed mucosa [9,21]. The features seen are dependent on the disease activity, which is commonly assessed using endoscopic scoring systems such as the Mayo Endoscopic Score [22] or the Ulcerative Colitis Endoscopic Index of Severity [23].

Histological diagnosis of ulcerative proctitis is based on widespread crypt architectural distortion, transmucosal inflammatory infiltrate with basal plasmacytosis [24,25] associated with active inflammation causing cryptitis and crypt abscesses. These features can take several weeks to be present, with basal plasmacytosis being the earliest diagnostic feature with the highest predictive value for the diagnosis of ulcerative colitis [26].

The first-line treatment of ulcerative proctitis is topical 5-aminosalicylates followed by oral preparations and corticosteroids [27]. However, despite initially presenting with distal disease, proximal extension has been reported at 17.8% at five years, and 31% at ten years [28,29,30]. Failure to respond to conventional treatment is seen in around one-third of patients [31], who subsequently require treatment with an advanced therapy (monoclonal antibody or small molecule inhibitor). Prospective data on these treatments are lacking as this symptomatic subgroup is usually excluded from clinical trials [32,33].

### 2.2. Crohn’s Disease

CD is a chronic inflammatory condition that can affect any part of the gastrointestinal tract, from the oral cavity to the anus. Most commonly, CD affects the terminal ileum and proximal colon and is characterised by a discontinuous, transmural inflammation [10]. Like UC, CD has an insidious onset, and it can take several months to years before a diagnosis is made [19]. The clinical presentation varies depending on the disease location and phenotype, which can include inflammatory stricturing, and/or penetrating behaviour leading to fistulation and intra-abdominal abscess formation [34]. The cardinal symptoms are chronic diarrhoea [35], abdominal pain, and weight loss [35]. Rectal bleeding may be seen in 45–50% of patients with Crohn’s colitis, but it is less commonly reported overall than in UC [36,37].

Isolated proctitis is an uncommon presentation of CD [38] as the rectum is often completely or partially spared [39]. Involvement of the rectum is associated with confluent involvement of the colon and/or small bowel, or is more commonly associated with perianal CD, with skin tags, fissures, fistulas, or abscesses. Perianal complications of CD affect up to 20% of patients [40,41,42].

Endoscopic features of CD are typically a patchy distribution of inflammation with small aphthous ulcers separated by normal mucosa that can enlarge to form deep serpiginous or linear ulcers with overhanging oedematous edges; these ulcers are separated by non-ulcerated mucosa which gives the classical “cobblestone” appearance. Fibro-stenotic strictures, fistulas, and abscesses can form at the site of transmural inflammation [20,39,43].

Histological features of CD are focal and discontinuous chronic inflammation, focal crypt irregularity, crypt atrophy, and granulomas. The presence of one of the features alone is not diagnostic and requires the right clinical context. These features may not be present for several weeks or months from the onset of symptoms [25,26,44].

## 3. Infective Proctitis

Infective proctitis is inflammation of the rectum, typically caused by sexually transmitted infections (STI). Infective proctitis predominately occurs in patients with a history of receptive anal sex leading to direct inoculation. Infections predominately affect the gay and bisexual male communities (MSM) and transgender women. Other risk factors for the development of infective proctitis include HIV seropositive status, sexually transmitted infections in the previous six months, unprotected receptive anal intercourse, and traumatic sex [1]. The most commonly isolated organisms causing infective proctitis are *Neisseria gonorrhoeae* (NG), *Chlamydia trachomatis* (CT, lymphogranuloma venereum (LGV) and non-lymphogranuloma venereum serovars), *Treponema pallidum*, herpes simplex virus (HSV), *Mycoplasma genitalium* (MG), and Mpox [45]. The causative organisms each have their own unique clinical features and endoscopic appearance (Figure 1), diagnostic test, and treatment.

### 3.1. Neisseria gonorrhoeae

*Neisseria gonorrhoeae* is a Gram-positive intracellular diplococcus. It is the second most common STI worldwide, causing urethritis in men and urethritis or cervicitis in women. It also affects extragenital sites, including the anorectum, and is the most common cause of infective proctitis worldwide, accounting for up to 30% of cases [46].

Anorectal NG has a reported prevalence of between 0 and 3% amongst women and 6 and 21% in MSM communities [47]. The rate amongst heterosexual men is unknown [48]. Multiple concurrent sites of infection are common, with isolated anorectal NG only found in 4% of women, compared to 70% of MSM [49]. Rectal NG is asymptomatic in up to 50% of male patients and over 95% of infected female patients [48,50].

The most common symptom in symptomatic men [51,52] is anal pain (67–87%), followed by rectal bleeding (33–45%) and purulent discharge (23–57%). The median time from symptom onset to first healthcare contact is 13 days.

Endoscopic examination demonstrates mucopurulent discharge with non-specific features such as erythema and oedema [53,54], and ulceration is rarely seen.

The diagnostic test of choice for rectal NG is nucleic acid amplification testing (NAAT) via a rectal swab, which has replaced microscopy. NAAT is more sensitive than microscopy [55], produces quicker results, and has the advantage that viable organisms are not required for detection. NAAT is currently unable to detect antimicrobial resistance, and with the emergence of multi-drug-resistant NG, culture has become increasingly important, with current guidance recommending specimens are sent for culture alongside NAAT [1].

NG has developed resistance to all antimicrobials used, including ceftriaxone, which threatens the last line of treatment [56]. UK guidelines currently recommend monotherapy with a single ceftriaxone dose, where sensitivities are not known, and with ciprofloxacin where NG is known to be sensitive [57]. European guidelines currently recommend dual antibiotic therapy with ceftriaxone and either azithromycin or doxycycline [58]. 

**Figure 1 microorganisms-12-02395-f001:**
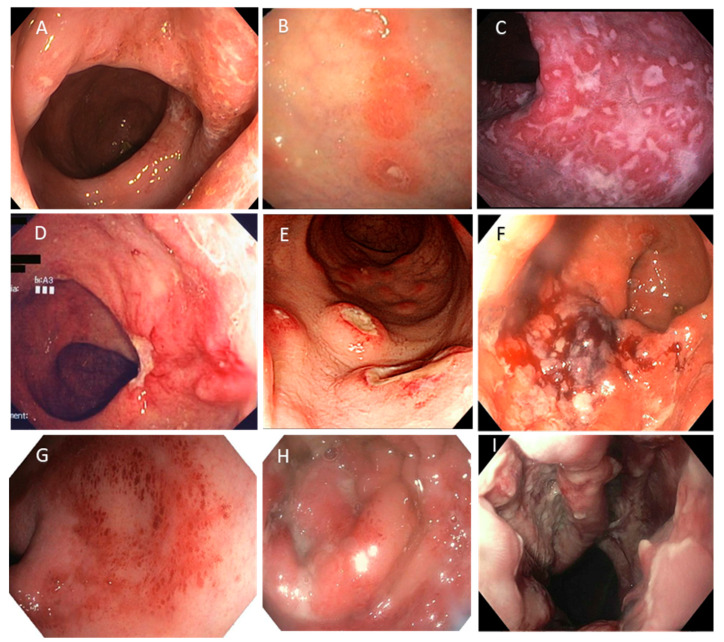
Endoscopic features of infective proctitis. (**A**,**B**)**:**
*Neisseria gonorrhoeae* causing mucosal oedema, erythema, white plaques, (**A**) and superficial erosions (**B**). (**C**,**D**): *Chlamydia trachomatis* lymphogranuloma venereum serovars causing multiple nodules with central ulceration with surrounding oedema and erythema, (**C**) and a large ulcer in the distal rectum with surrounding friable mucosa (**D**). (**E**,**F**): *Treponema pallidum* causing ulceration with polypoid hyperaemic lesions (**E**) and a friable ulcerated rectal mass (**F**). (**G**,**H**): Herpes simplex virus causing erythematous and friable mucosa (**G**) with ulceration and mucopurulent discharge (**H**). (**I**): Mpox causing severe proctitis with ulceration with pustular lesions. All images reproduced and adapted with permissions. All copyrights remain with the original copyright holders. (**A**): Sigle and Kim [53] © Georg Thieme Verlag KG with permission. (**B**): Coelho et al. [46] ©Hellenic Society of Gastroenterology. Licence under CC-BY-NC 4.0. (**C**): Law et al. [59] © 2020, the AGA Institute with permission from Elsevier. (**D**): Di Altobrando et al. [60] © Crohn’s & Colitis Foundation. Licence under CC-BY-NC 4.0. (**E**): You et al. [61] © The Korean Journal of Gastroenterology. Licence under CC BY-NC 3.0. (**F**): Costales-Cantrell et al. [62] © 2021 Society of General Internal Medicine, with permission from Springer Nature. (**G**,**H**): Sandgren et al. [63] © 2017. Licence under CC-BY 4.0. (**I**): Mavilia et al. [64] © 2023 the AGA Institute, with permission from Elsevier.

### 3.2. Chlamydia trachomatis

*Chlamydia trachomatis* is an obligate intracellular Gram-negative coccus that has multiple different serovars. Serovars D-K cause non-LGV infections; these are limited to epithelial surfaces and are non-invasive. Serovars L1-3, which are invasive serovars, cause LGV infections. CT infects the epithelial cells of the oropharynx, urogenital tract, and/or gastrointestinal tract. CT is the most commonly diagnosed STI worldwide and the second most common cause of infective proctitis, with a rising incidence [45]. The positive rates of rectal CT have been reported to be between 10 and 15% amongst MSM communities [65] and 5 and 9% in women [65].

Most rectal CT infections are asymptomatic, with 15–20% of cases causing symptomatic proctitis. Among MSM communities, the primary route of infection is via receptive anal sex, although associations with other sexual practices have been demonstrated [66]. In women, the rate of rectal CT is similar between those who do, and do not, report anal sex [67]; therefore, cross-over infections due to the close proximity of the vagina and anus alongside hygiene practices are thought likely to contribute [68].

#### 3.2.1. Non-LGV *Chlamydia trachomatis*

Non-LGV rectal serovars are asymptomatic in a large majority of cases; in the small proportion of cases that develop proctitis, this tends to be mild, with endoscopy showing non-specific inflammation, erythema, and injected mucosa but rarely ulceration or abscesses [53,69,70]. The diagnostic test of choice is NAAT collected from a rectal swab [1] either via proctoscopy or taken blind by the patient [71]. Current guidelines suggest non-LGV rectal CT infections should be treated with doxycycline, 100 mg twice a day for 7 days, or, alternatively, azithromycin, 1 g as a single dose [45,72].

#### 3.2.2. Lymphogranuloma Venereum *Chlamydia trachomatis*

Lymphogranuloma venereum is caused by one of the three invasive serovars of CT, L1, L2, or L3, with L2 being the most common strain, often subcategorized as L2b [73,74]. Historically, LGV was considered a tropical disease endemic in parts of Africa, Latin America, and Asia, but rare in Western countries. Since an outbreak in the Netherlands in 2003 [75], it has become endemic in Europe, with infection rates rising steadily. LGV now causes 10–25% [76,77,78] of rectal CT infections.

LGV proctitis is almost exclusive to men, and specifically MSM, representing up to 99% of cases; the small number of cases identified in women are thought to be of transgender women [79]. Initially, there was a strong association with HIV co-infection, with HIV present in >75% of patients diagnosed with LGV [80]. There has since been a steady increase in the number of LGV cases in HIV-negative men [81], who now represent the majority of cases in France [78], the UK [82], Belgium, [83], and Italy [73]. The increase in cases among HIV-negative men has corresponded with the rollout of HIV PrEP.

Amongst heterosexuals, where the primary inoculation site is via the penis or vagina, inguinal and femoral lymphadenitis develops, typically unilaterally, and as the disease progresses, the lymph nodes become matted and form bubos. Due to the severe inflammatory reaction, these bubos ulcerate, causing chronic fistulae, which extend into other structures, including the rectum, leading to proctitis, peri-rectal abscesses, and rectal strictures [84]. This can mimic perianal CD. The prevalence of genital LGV is rare amongst MSM [85]. When the rectum is the primary site of inoculation, proctitis is the most common manifestation, representing up to 96% of all presentations [86]. Symptoms include anal pain, anorectal bleeding, mucoid rectal discharge, tenesmus, and constipation [84]. Rectal discharge, rectal bleeding, and anal pain are the most-reported symptoms [75,80,87,88], whereas inguinal lymphadenopathy and penile lesions are rarely seen. In a prospective multi-centre case-controlled study of patients presenting with symptomatic LGV [88], tenesmus, constipation, and anal discharge were up to seven times more likely to be reported in the LGV-positive cases; however, they were less frequently reported overall than other symptoms. The asymptomatic rate has been reported at 27–45% [89,90]. Left untreated, LGV proctitis can result in rectal strictures [91], fistulas, and masses that can be mistaken for rectal cancers [92,93].

Endoscopically, features of LGV proctitis are non-specific, with significant overlap with IBD. During the initial outbreak, Nieuwenhuis et al. [75] reported 12 patients who underwent proctosigmoidoscopy. Mucopurulent exudates and ulceration were common findings, alongside erythema and friable mucosa, which were less common. One patient had a tumour-like mass, and three patients had extrinsic compression reported. Given the similarities with IBD, misdiagnosis is commonly reported. Soni et al. [87] identified 12 cases of LGV who were initially misdiagnosed with IBD. Endoscopic features ranged from normal mucosa to moderately active proctitis and ulceration; in only one case did the findings extend into the sigmoid. Histologically, cryptitis and crypt abscess were common histological findings, but distortion of the crypt architecture was rare. Granulomas were seen in five patients, and ulceration was seen in seven patients. Di Altobrando et al. [60] reported on 11 patients diagnosed with LGV; of these, 8 had previously been treated unsuccessfully for IBD. Rectal biopsies demonstrated lymphoplasmacytic infiltrates, lymphohistiocytic colitis, cryptitis with focal crypt distortion, and crypt abscesses interpreted as Crohn’s disease.

Testing for LGV should be carried out on all MSM with positive CT rectal swabs, contacts of confirmed cases, or symptoms consistent with LGV [94]. Current guidance [84] suggests that testing is done in two stages. Firstly, a commercially available NAAT should be used to detect CT DNA/RNA. If CT is detected, then an LGV genovar-specific CT NAAT should be used, although this has not yet been fully evaluated in rectal LGV. Alternatives are chlamydia genus-specific serological assays. Treatment of LGV is recommended with doxycycline, 100 mg twice a day for 21 days; alternative antibiotics include minocycline, azithromycin, or erythromycin [84,94].

### 3.3. Treponema pallidum

Syphilis is caused by the spirochete *Treponema pallidum*. Over recent years, the incidence of syphilis has been increasing, particularly in the MSM community [95]. Primary syphilis usually appears 10–90 days after direct contact with an infective skin lesion, typically a painless ulcerative and indurated lesion with a clean base (chancre), often in the anogenital area [96]. If not identified and treated, it can progress to secondary syphilis, which has many manifestations, including a palmoplantar rash, oral ulceration, proctitis, and condylomata lata; however, it can affect almost any system in the body. Tertiary infection occurs several years after infection and results in major neurological or cardiovascular sequelae.

Syphilis proctitis is a rare presentation of syphilis infection and only causes 1% of infective proctitis overall [97,98]. Syphilis proctitis can present with a range of symptoms. Arnold et al. [99] reported on seven patients with syphilis proctitis; rectal bleeding and rectal pain were the most common symptoms. There are several case reports of syphilis proctitis presenting with rectal masses often mistaken for rectal cancers [100,101,102,103]. A recent literature review [104] identified 61 cases of published lower GI syphilis. The most commonly reported symptoms were haematochezia (67%), followed by rectal pain (46%), abdominal pain (28%), tenesmus (25%), mucous discharge (23%), diarrhoea (23%), and constipation (13%). Endoscopically, 42% demonstrated a rectal mass and 35% demonstrated anorectal ulceration, described as "atypical" in appearance [105]. Fissures, fistulas, and abscesses were also reported but were rare. Chronic lymphoplasmacytic infiltration was the most common histopathological finding (75%), followed by acute inflammation/cryptitis/ crypt abscess (46%), ulceration (22%), and granulomas (22%).

The presumptive diagnosis of syphilis is made with serological tests, including non-treponemal and treponemal tests. Confirmatory tests are required, given the potential for false positives or PCR results of ulcers or rectal biopsies [46,96].

The treatment of syphilis is with a single intramuscular injection of penicillin G benzathine, 2.4 million units; alternative antibiotics are doxycycline or ceftriaxone [96,106].

### 3.4. Herpes Simplex Virus

Traditionally, HSV-1 caused fever and perioral cold sores, and HSV-2 was associated with painful blisters in the anogenital area. However, in the modern age, both types of HSV can be found in each location. HSV is transmitted by intimate personal contact, and, consequently, HSV proctitis is usually transmitted by unprotected anal sex or oral sex, particularly in the MSM community. HSV is the third most isolated pathogen amongst MSM presenting with proctitis symptoms [97], and the most common pathogen amongst HIV-positive men [98].

HSV presents with a vesicular eruption near mucocutaneous junctions. HSV proctitis is rarely asymptomatic [107], with anorectal pain, often severe, reported in up to 87% of cases [108]. Amongst men presenting with external anal ulceration, HSV was isolated in 83% of cases [98]; however, only 31–33% of men with HSV proctitis have external ulceration. Other symptoms include rectal discharge, tenesmus, and/or rectal bleeding. Difficulty urinating, sacral paraesthesia, and faecal incontinence have also been reported [53].

Endoscopic features are confined to the distal rectum and include friable mucosa, diffuse distal ulceration, and vesicular lesions [63,109,110,111].

Diagnosis is made with PCR swabs of rectal mucosa. Treatment is with acyclovir, valaciclovir, or famciclovir [112]. In HIV-positive men, empirical treatment should be considered if patients present with severe anal pain [1].

### 3.5. Mycoplasma genitalium

*Mycoplasma genitalium* is a common urogenital bacterial STI amongst both men and women, typically causing urethritis. Rectal MG is usually asymptomatic; however, positive swabs have been reported in patients with symptoms of proctitis, particularly amongst MSM. The prevalence rates of positive rectal swabs among MSM populations have been reported at 4–9.5% [113,114,115]. The prevalence amongst women is less reported, with a positive rate of 22% amongst high-risk women [116], although no associations with symptoms of proctitis were demonstrated. Bissessor et al. [117] reported on 18 men with MG proctitis, and of these, 7 presented with anal discharge, 4 with rectal pain, and 7 with both.

The role of anorectal MG in MSM presenting with proctitis is debated. MG has been reported as the sole pathogen in 12–17% of MSM presenting with proctitis [117,118]. However, multiple studies have found no correlation between a positive MG rectal swab and symptomology [113,114,119]; furthermore, co-infection with other bacterial infections is very common [114,120]. Endoscopic features are non-specific, with erythema and erosions with mucopurulent exudate [46].

Recent guidelines [121] suggest that rectal testing with NAAT is only indicated if CT and NG have been excluded. Macrolide-resistant strains have been reported in as many as 84% of cases in some studies [114], meaning a positive test should be followed up with an assay capable of detecting macrolide resistance.

Current UK guidelines [122] recommend treatment with 7 days of doxycycline followed by a short course of azithromycin. If resistance is suspected or proven, then moxifloxacin should be used. European guidelines do not recommend doxycycline as a first-line treatment [121].

### 3.6. Mpox

Mpox is caused by the Monkey pox virus, a member of the *Poxviridae* family and *Orthopoxvirus* genus. Historically, Mpox was endemic to central and west Africa; however, it emerged as a global concern in May 2022, as community transmission was reported worldwide. The 2022 outbreak had over 100,000 confirmed cases. In August 2024, it was designated a public health emergency of international concern by the World Health Organization [123].

During the 2022 outbreak, over 98% of cases were identified in men, and over 95% of cases occurred in MSM [124,125,126]. The most common symptoms identified were prodromal symptoms, including fever and lymphadenopathy, followed by a rash present in >99% of cases. The rash is typically umbilicated, vesicular, and/or pustular with an erythematous halo [4,124,125]; it is common for there to be under 5 lesions, and there are rarely more than 20 lesions [126]. The lesions tend to manifest anogenitally but can be present on any part of the body [125,126].

Proctitis was identified early in the outbreak as an additional symptom of Mpox [127], with 14–39% of patients reporting symptoms of proctitis [4,124,125,126,128]. The most commonly reported rectal symptoms [129] were rectal pain (86%), rectal discharge (38%), and painful defecation (27%). Tarin-Vincent et al. [4] reported that MSM who had receptive anal sex were more likely to report proctitis than those who did not and were more likely to have systemic features before the appearance of a rash. There are reports [129] of patients presenting with proctitis prior to the development of a skin rash and, in rare instances, proctitis as the only manifestation of Mpox [129]. A case series of Mpox-positive women reported that proctitis was more common in transwomen (54%) than cis-women (11%) [130].

Specific endoscopic features of Mpox remain limited; Mavilia et al. [64] published a single case reporting sigmoidoscopy demonstrating severe proctitis with deep ulcerations and scattered pustular lesions. Mazzotta et al. [131] reported on four patients with endoscopy showing oedematous, erythematous, and friable mucosa with small erosive ulceration. A solitary rectal ulcer was seen in one patient.

Diagnosis of Mpox is with PCR testing of the Mpox DNA taken from a swab of skin or anorectal lesions. Treatment for Mpox for most patients is supportive, such as non-steroidal anti-inflammatories, stool softeners, paracetamol, and topical lidocaine. In immunocompromised patients, or in those with severe manifestations or severe pain or patients at risk of severe complications, tecovirimat can be used [132]. Causes of infective proctitis are found in Table 2.

## 4. Distinguishing Inflammatory Bowel Disease and Infective Proctitis

There are many overlapping features between infective and inflammatory proctitis, and, consequently, a high index of suspicion is required. Many of the distinguishing features rely on a detailed, specific, and accurate clinical history, as outlined in Table 3. Eliciting the nature and symptomology of the rectal syndrome and establishing high-risk features are crucial to avoid misdiagnosis. This includes a detailed sexual history, which is often poorly done in a non-STI clinic setting. Acute symptom onset, anorectal pain, and purulent discharge are very common features in infective causes, whereas an indolent onset with bloody diarrhoea is more suggestive of IBD.

In a case-controlled series comparing 10 UC, 10 CD, and 10 IP patients [133], the authors found that the only significant differences in the clinical presentation were an HIV-positive status, receptive anal sex, rectal pain, and rectal discharge; all other symptoms were non-discriminatory. Having co-existing HIV and IBD is rare, and rates of IBD in the HIV-infected population are significantly below the rates in the general population [134]. Those who do have co-existing HIV and IBD rarely have severe IBD relapses, and it has been proposed that due to the rectosigmoid distribution, some patients may have initially been misdiagnosed [134].

There have been several case series identifying patients who were misdiagnosed with IBD before subsequent diagnosis of infective proctitis [8,60,87,135,136,137,138,139]. In all reports, the misdiagnosed patients were men, predominately MSM, with a high HIV seropositive rate, and a sexual history of recent receptive anal intercourse. Alongside these risk factors, many patients reported having symptoms more suggestive of an acute infective cause. Typically, patients in the series were treated for several months (range 1–36 months) without any notable response. All responded to treatment with antibiotics after a subsequent diagnosis. There are reports of patients undergoing surgery for presumed perianal CD [60] or receiving infliximab treatment [8] prior to the correct diagnosis. Failure of initial treatment should always lead a clinician to reconsider the original diagnosis. In UP, a failure of initial treatment is common, with up to 31% of patients failing conventional therapy [8,31]; however, in the presence of other risk factors, a high degree of clinical suspicion should be exercised. Identified patients who have risk factors alongside atypical features and/or have failed to respond to treatment should be investigated for infective causes; this is key to avoiding misdiagnosis; see Figure 2. The British association of sexual health and HIV recommend considering empirical treatment to cover LGV, NG, and HSV in patients with likely sexually transmitted proctitis [140].

Endoscopic and histological appearances in infective proctitis can appear indistinguishable from IBD. There are no specific endoscopic features that are diagnostic of UC or CD [141] or that distinguish IBD from IP [133]. However, in several misdiagnosed cases, lesions and inflammation were often identified as “non-specific” or “atypical”, suggesting a degree of doubt. To increase the likelihood of identifying subtle endoscopic features, adequate bowel preparation is required prior to the procedure, taking into account any pre-existing mucosal damage [142]. Levy et al. [8] reported on 16 patients incorrectly diagnosed with IBD and noted that endoscopic features were indistinguishable, with a varying degree of inflammation and rectal ulcers commonly present. Histopathological findings included cryptitis, crypt distortion, crypt abscesses, and granulomas. Soni et al. [87] reported on 12 patients with LGV incorrectly diagnosed who were treated for IBD; their endoscopic features ranged from mild proctitis to severe ulceration. Histology demonstrated cryptitis and crypt abscesses with granulomas present in five cases; however, a distortion of crypt architecture was not a prominent feature. The absence of crypt architectural distortion has previously been identified as a significant distinguishing feature [133], alongside raised Paneth cells and increased eosinophil levels, both of which are more common in IBD. These features are consistent with previously established histological differences between acute infective colitis and IBD, which include basal plasmacytosis, crypt distortion, crypt atrophy, epithelioid granulomas, and Paneth cell metaplasia, which are all strong predictors of IBD and rare in other acute infective colitis [143,144,145]. Histological features, which favour IBD over infective causes, take several weeks to be present [141], and, therefore, in an acute presentation, may be absent.

## 5. Conclusions

With many similar and overlapping clinical features and endoscopic and histopathological findings, infective proctitis can be indistinguishable from IBD. When patients present with proctitis, a detailed and specific clinical history is required that considers the high-risk features for sexually transmitted infections, particularly in the MSM community. High-risk patients should have rectal swabs for *Chlamydia trachomatis* and *Neisseria gonorrhoeae* alongside syphilis serology. In the presence of other clinical features such as skin lesions, Mpox or HSV should be considered. In patients with risk factors and/or atypical features, failure to consider infective proctitis can lead to delayed diagnosis, the inadvertent initiation of immunosuppressive treatment, and complications that could be avoided.

## Figures and Tables

**Figure 2 microorganisms-12-02395-f002:**
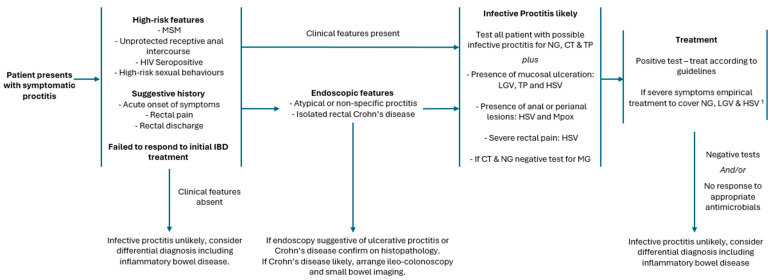
Suggested flow chart to distinguish infective proctitis from inflammatory proctitis. MSM, men who have sex with men; IBD, inflammatory bowel disease; NG, *Neisseria gonorrhoeae*; CT, *Chlamydia trachomatis*; LGV, Lymphogranuloma venereum; TP, *Treponema pallidum*; HSV, herpes simplex virus; MG, *Mycoplasma genitalium.*
^1^ UK guidelines recommend empirical treatment to cover NG, CT (including LGV) and HSV for patient with and probable sexually transmitted proctitis and severe symptoms (Richardson et al. [140]).

**Table 1 microorganisms-12-02395-t001:** Risk factors for infective proctitis.

Risk Factors for Infective Proctitis
MSM or transgender women
HIV seropositive status
Unprotected receptive anal intercourse
Other sexually transmitted infection in previous six months
High-risk sexual behaviours
Traumatic sex
Multiple sexual partners
Group sex
Chemsex ^1^

Adapted from de Vries et al. [1]. MSM; men who have sex with men, referring to gay or bisexual men. ^1^ Chemsex refers to sexualized drug use (commonly methamphetamines, mephedrone, gamma hydroxybutyrate (GHB), and gamma-butyrolactone (GBL)).

**Table 2 microorganisms-12-02395-t002:** Causes of infective proctitis, its common presenting symptoms, endoscopic features, diagnostic tests, and first-line treatments.

	Common Clinical Features	Endoscopic Features	Diagnostic Test	First-Line Treatment
*Neisseria gonorrhoeae*	Rectal pain, rectal bleeding, purulent discharge, tenesmus.	Purulent discharge, erythema, and loss of vascular pattern. Ulceration is not common.	NAAT via rectal swab or tissue sampling. Culture to assess antibiotic resistance.	Ceftriaxone, 1 g IM once if sensitivities are unknown. Ciprofloxacin if known to be sensitive [57].
*Chlamydia trachomatis* serovars D-K	Usually asymptomatic. Rectal pain, tenesmus, mucopurulent or bloody discharge.	Mild inflammation with erythema, friability, and erosions. Ulceration is rare.	NAAT via rectal swab or tissue sampling.	Doxycycline, 100 mg PO BD for 7 days or azithromycin, 1 g PO as a single dose [72].
*Chlamydia trachomatis* serovars L1-3 (LGV)	Rectal pain, mucopurulent discharge, anorectal bleeding, tenesmus, and constipation.	Mucopurulent exudate and ulceration. Erythematous and friable mucosa. Fistulas, strictures, abscesses, and masses can be seen.	NAAT for CT via rectal swab followed by LGV-specific NAAT.	Doxycycline, 100 mg BD for 21 days [84].
*Treponema pallidum*	Rectal bleeding, rectal pain, abdominal pain, tenesmus, diarrhoea, mucous discharge.	Anorectal ulceration, rectal masses. Fissures, fistulas, and abscesses can be present.	Non-treponemal and treponemal serology. Tissue biopsy with staining.	Penicillin G benzathine, 2.4 million units IM, single dose [106].
Herpes simplex virus	Severe rectal pain, tenesmus, constipation, rectal discharge, perianal ulceration, sacral paraesthesia.	Vesicular lesions, mucosal oedema, and ulceration. Confined to distal rectum.	NAAT via rectal swab or biopsy.	Acyclovir, 400 mg TDS PO for 5 days, or valaciclovir 500 mg BD for 5 days [112].
*Mycoplasma genitalium*	Rectal pain and rectal discharge.	Non-specific erythema, erosions.	NAAT via rectal swab, only if NG and CT are excluded.	Doxycycline, 100 mg BD PO for 7 days followed by azithromycin, 1 g PO once, followed by 500 mg PO OD for 2 days. If known macrolide resistance, moxifloxacin, 400 mg OD PO for 7 days [113].
Mpox	Prodromal fever and lymphadenopathy. Rash. Rectal pain, mucopurulent discharge, and painful defecation.	Oedematous, erythematous, and friable mucosa with ulceration.	NAAT via skin lesion or rectal swab.	Symptomatic management. In severe cases, tecovirimat, 600 mg BD for 14 days [132].

LGV, lymphogranuloma venereum; NAAT, nucleic acid amplification testing; NG, *Neisseria gonorrhoeae*; CT, *Chlamydia trachomatis*; IM, intramuscular; PO, by mouth; BD, twice a day; TDS, three times a day.

**Table 3 microorganisms-12-02395-t003:** Distinguishing features between infective proctitis and inflammatory proctitis.

	Inflammatory Proctitis	Infective Proctitis
Biological sex	Male = Female	Predominately male
Sexuality	Any	Predominately gay or bisexual men, or transgender women
HIV seropositive status	Rare	Common
Recent unprotected receptive anal sex	Unrelated	Very common
Time from symptom onset to presentation	Weeks to several months	Days to short weeks
Rectal pain	Uncommon (common in perianal Crohn’s disease)	Very common
Mucopurulent discharge	Uncommon	Common
Diagnostic test	No	Yes
Improves with antimicrobials	No	Yes

## Data Availability

No new data were created or analyzed in this study. Data sharing is not applicable to this article.

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
