# Peer review of "Separating Infectious Proctitis from Inflammatory Bowel Disease—A Common Clinical Conundrum"

_microorganisms, 2024, doi:10.3390/microorganisms12122395_

Round 1
Reviewer 1 Report
Comments and Suggestions for Authors
Separating Infectious Proctitis from Inflammatory Bowel Disease – A common clinical Conundrum.
Distinguishing infective proctitis from inflammatory bowel disease remains a significant clinical challenge as there is significant overlap in the clinical presentation and their endoscopic and histological features. The review compares and highlights the distinguishing hallmarks of both inflammatory and infective causes of proctitis. It provides a practical guide to highlight the key features that clinicians should focus on both clinical and diagnostic to avoid potential misdiagnosi. The review is well prepared and well written. Very easy to follow.
No comments
Author Response
Many thanks for taking time to review the manuscript and your positive feedback.
Reviewer 2 Report
Comments and Suggestions for Authors
I read with particular interest this excellent review. I believe the distinction between ulcerative proctitis and infectious proctitis is clinically very useful for us endoscopists.
The figures are truly helpful, as are the flowcharts provided.
A few minor comments:
- In the legend of Figure 1, I could not find any comments on panel D.
- I would add an initial preliminary comment highlighting that, since these conditions are represented by different nuances of the mucosal endoscopic appearance, adequate bowel preparation is required before the endoscopic examination. However, this preparation should take into account any pre-existing mucosal damage (I recommend discussing the most recent review on the subject: https://pubmed.ncbi.nlm.nih.gov/37034970/).
Author Response
Many thanks for taking time to read our review and provide such positive and constructive feedback.
"- In the legend of Figure 1, I could not find any comments on panel D."
Thank you for pointing this out. We have changed figure 1 by removing and replacing a few photos due to copyright restrictions. All the photos are now referenced in the Legend and there is also an attached copyright statement.
"- I would add an initial preliminary comment highlighting that, since these conditions are represented by different nuances of the mucosal endoscopic appearance..."
We agree that this is an excellent consideration to make when attempting to spot and diagnose subtle changes. We have now commented upon this on page 12 .